# Problematic Internet Use, Non-Medical Use of Prescription Drugs, and Depressive Symptoms among Adolescents: A Large-Scale Study in China

**DOI:** 10.3390/ijerph17030774

**Published:** 2020-01-26

**Authors:** Beifang Fan, Wanxing Wang, Tian Wang, Bo Xie, Huimin Zhang, Yuhua Liao, Ciyong Lu, Lan Guo

**Affiliations:** 1Department of Psychiatry, Shenzhen Nanshan Center for Chronic Disease Control, Shenzhen 518000, China; fanbf@foxmail.com (B.F.); sznsxb@163.com (B.X.); zhangzhanghuimin@163.com (H.Z.); liaoyuhua0011@163.com (Y.L.); 2Department of Medical Statistics and Epidemiology, School of Public Health, Sun Yat-sen University, Guangzhou 510080, China; wgg0808@163.com (W.W.); wangt97@mail2.sysu.edu.cn (T.W.); luciyong@mail.sysu.edu.cn (C.L.); 3Guangdong Engineering Technology Research Center of Nutrition Translation, Guangzhou, 510080, China

**Keywords:** problematic Internet use, non-medical use of prescription drugs, depressive symptoms, adolescents

## Abstract

This large-scale study aimed to test, among Chinese adolescents, the association between problematic Internet use (PIU), non-medical use of prescription drugs (NMUPD), and depressive symptoms, as well as the mediating effects of NMUPD on the associations above. This study used the data from the 2017 National School-based Chinese Adolescents Health Survey, and 24,345 students’ questionnaires qualified for the analyses. Generalized linear mixed models and path models were performed. In the models without mediation, PIU was associated with depressive symptoms (unstandardized β estimate = 0.26, 95% CI = 0.25–0.27); frequent use of opioid or sedative was also related to depressive symptoms (unstandardized β estimate for opioid = 2.77, 95% CI = 1.90–3.63; unstandardized β estimate for sedative = 4.45, 95% CI = 3.02–5.88). Additionally, the results of the path models indicated that opioid misuse partially mediated the association between PIU and depressive symptoms. PIU and opioid/sedative misuse were related to the increased risk of depressive symptoms, respectively. The association above might be complicated, and PIU may elevate the risk of opioid or sedative misuse and depressive symptoms, which in turn could worsen the situation of PIU and vice versa. Multidisciplinary health intervention programs to prevent adolescents involving in PIU, as well as NMPUD, are recommended to be provided.

## 1. Introduction

Internet use has proliferated dramatically over the last decades and has become an internal part of contemporary life [1]. Excessive or maladaptive use of the Internet has been termed as “problematic Internet use (PIU)” or “Internet addiction”, which has been reported to be related to a series of psychosocial impairments and substance-use problems (e.g., depression and drug use) [2,3]. Recently, Internet gaming disorder has been included in the section on addictive disorders in the 11th Revision of the International Classification of Diseases (ICD-11) as a pattern of behavior characterized by impaired control over gaming, indicating that PIU may develop into a severe social problem [4]. Adolescence represents a vulnerable developmental stage between childhood to adulthood, which is characterized by increased levels of novelty seeking and exploration along with a wide range of risk-taking behaviors or mental health problems [5]. The report of the World Health Organization (WHO) showed that adolescent PIU had been a public health concern worldwide, and China is no exception [6]. Considering the brain undergoes rapid development during adolescence, adolescents involved in PIU may be more vulnerable to psychological disorders, such as depressive symptoms.

Depressive symptoms are one of the most common mental disorders among adolescents. Evidence suggests that adolescent depressive symptoms can not only have long-lasting influences on the development of cognitive abilities and social skills into adulthood, but also may lead to clinical depression, placing a heavy financial burden on individuals, families, and society [7]. Previous studies in Western countries reported that approximately 20–50% of adolescents had suffered from long-term depressive symptoms [8], and our prior study in China also showed that about 6.4% of high school students reported having depressive symptoms with suicidal ideation [9]. Although the association between PIU and depressive symptoms has been documented in prior studies [10,11], the influence path of PIU on depressive symptoms is still unclear.

Over the last two decades, the rapid growth of non-medical use of prescription drugs (NMUPD) among adolescents has drawn increased attention [12]. According to the report of the 2014 National Survey on Drug Use and Health (NSDUH) in the United States, prescription drugs were the second most popular type of drugs among adolescents [13]. A recent study also demonstrated that 2.0% of Chinese high school students reported frequent use of opioids, and 1.8% admitted frequent use of sedatives [14]. Previous studies demonstrated that PIU was reported to be associated with substance use, and PIU and substance use may share similar biological characteristics (e.g., similar vulnerable brain regions) [3,15]. Moreover, substance use may affect an individual’s brain function, as well as their ability to self-regulate, leading to the development of depressive symptoms [16,17]. Therefore, NMUPD may play a mediator role between PIU and depressive symptoms. First, Caplan’s theory proposed that a preference for online social interaction and the use of the Internet for mood regulation can predict deficient self-regulation of Internet use (i.e., PIU), which was a significant predictor of depressive symptoms [18]. Besides, based on the cognitive–behavioral theory of generalized PIU introduced by Davis, depressive symptoms might predispose individuals to develop maladaptive Internet-related cognitions and behaviors that ultimately result in adverse outcomes (e.g., substance use or PIU) [19,20]. Then, there might be two mediation models (incorporating NMUPD as the mediators) in the association between PIU and depressive symptoms from two orders: (1) PIU—NMUPD—depressive symptoms; and (2) depressive symptoms—NMUPD—PIU.

With the modernization of society, Chinese adolescents are more easily exposed to PIU, NMUPD, and depressive symptoms. However, there is a lack of studies in China considering the effects of NMUPD on the association between PIU and depressive symptoms. Therefore, we conducted this large-scale study aimed to test, among Chinese adolescents, the association between PIU, NMUPD, and depressive symptoms, as well as the mediating effects of NMUPD on the association between PIU and depressive symptoms.

## 2. Materials and Methods

### 2.1. Study Design and Participants

This study adopted the data from the 2017 National School-based Chinese Adolescents Health Survey (SCAHS), which is an ongoing study about the health risk behaviors and mental health problems among Chinese adolescents (7th–12th grade) conducted by our group. SCAHS performed a series of large-scale cross-sectional surveys every two years since 2007. The 2017 SCAHS utilized a multi-stage stratified cluster random sampling method to recruit a representative sample of adolescents in Guangdong province, and the procedures for data collection have been described in detail in previous publications [21]. Briefly, Guangdong province was firstly categorized into three stratifications according to geographic locations (Yue Dong, Yue Xi, Yue Bei, and Pearl River Delta) and gross domestic product (GDP) per capita (high-level, middle-level, and low-level). Then, we randomly selected two representative cities from each stratification. Considering adolescence is often described as occurring between 13 and 18 years of age (roughly the period of high school for much of the world) [22], in the present study, four vocational high schools and four general high schools were selected from each chosen city. Finally, two classes were randomly selected from each grade within the chosen schools. All available students were invited to participate in this study voluntarily, and 24,345 students’ questionnaires were completed and qualified for analyses, resulting in a response rate of 92.1%. The anonymity of the questionnaires was guaranteed to elevate the validity of self-reports of stigmatized behaviors [23], and our research assistants administered these questionnaires during a regular class period without the presence of teachers (to avoid any potential information bias). Of the total sample, 51.5% (12,526) were boys, and the mean age of the students was 15.2 ± 1.8 years. The study procedures were carried out in accordance with the Declaration of Helsinki. The School of Public Health Institutional Review Board of the Sun Yat-sen University approved the study (the ethic code: L201720). All participants were informed about the study, and all provided written informed consents. If the participant was under 18 years of age, a written informed consent letter was obtained from one of the student’s parents (or legal guardian).

### 2.2. Measures

Depressive symptoms were estimated by the Center for Epidemiologic Studies Depression (CES-D) Scale proposed by Radloff (1977), which has been validated and widely used among Chinese adolescents with satisfactory psychometric properties [24], and the Cronbach’s alpha for this CES-D scale was 0.88 in the present study (the Cronbach’s alpha ranging from 0.85 to 0.90 in Radloff’s study). The total CES-D score ranges from 0 to 60, where higher scores indicate more significant depressive symptoms [25]. A cutoff score of 28 points was utilized to identify students at risk for clinical depression, also calling having depressive symptoms. This cutoff score has been used in previous studies in Chinese adolescents [26,27].

PIU was measured by Young’s Internet Addiction Test (IAT) proposed by Young (1998), which has been validated and utilized among Chinese adolescents with satisfactory psychometric properties [28,29], and the Cronbach’s alpha for IAT was 0.91 in this study. The IAT includes 20 items rated on a five-point Likert scale (from 1 = not at all to 5 = always) [30], where the total IAT score ranges from 20 to 100 and a higher score representing a greater level of inclination to PIU.

In the present study, NMUPD consists of non-medical use of opioids and sedatives. This list of opioids and sedatives was developed according to the report of the Guangdong Food and Drug Administration, with a focus on the prescription medications that have been widely used by adolescent drug abusers in rehabilitation centers of China. Opioids consist of compounded cough syrup with codeine (codeine), compounded licorice tablets (opium), tramadol hydrochloride, and diphenoxylate. Sedatives included diazepam or triazolam (benzodiazepines), compounded aminopyrine phenacetine tablets (barbiturates), and scopolamine hydrobromide tablets (barbiturates). Non-medical use of opioids or sedatives was measured by asking students how many times have you used the prescription medications as mentioned above for a non-medical purpose in the past year, and response options included “never”, “once or twice”, and “at least 3 times”. Students who selected “never” were considered as abstainers, those who answered “once or twice” were thought as experimenters, and those who selected “at least three times” were treated as frequent users [30].

Information on sociodemographic variables were also collected, including gender (1 = boy, 2 = girl), age, living arrangement (responses were coded as “living in a two-parent family” = 1, “living in a single-parent family” = 2, and “living with others” = 3), household socioeconomic status (HSS, available responses were “above average” = 1, “average” = 2, and “below average” = 3), academic performance (responses were also coded as “above average” = 1, “average” = 2, and “below average” = 3). Family relationships, classmate relations, and relationships with teachers were assessed by asking how students perceived their relationships with family members, classmates, and teachers (available responses included “good” = 1, “average” = 2, and “poor” = 3).

### 2.3. Statistical Analysis

First, descriptive analyses were used to describe sample characteristics, and the *t*-tests and one-way ANOVA tests were performed to compare the differences in the CES-D scores. Descriptive analyses stratified by depressive symptoms were also conducted, and the chi-square tests or *t*-tests were used to describe the distribution of depressive symptoms in categorical or continuous variables. Second, considering this study used a multi-stage sampling design in which students were clustered into classes, generalized linear mixed models were fitted in which classes were treated as groups, and unstandardized β coefficients were reported. Univariable generalized linear mixed models were first conducted to estimate the potential association of PIU or NMUPD with depressive symptoms. Multivariable generalized linear mixed models, in which variables that were significant at the 0.10 level in the univariate analyses or widely reported in the literature were simultaneously incorporated, were performed to test the independent association of PIU or NMUPD with depressive symptoms. Third, multiplicative interaction items were tested by entering a cross-product term for opioid misuse or sedative misuse and PIU along with the main effect terms for each to the multivariable generalized linear mixed models, and *p*-values for the multiplicative interaction were calculated. Fourth, path models utilizing the maximum likelihood approach were conducted to assess the mediating effects of opioid or sedative misuse on the association between PIU and depressive symptoms. We first ordered the variables as follows: PIU—opioid or sedative misuse—depressive symptoms, and then an alternative model with a different order of the variables were assessed: depressive symptoms—opioid or sedative misuse—PIU. Due to the variables of the CES-D and IAT scores being continuous variables, and the measures of opioid and sedative misuse were categorized, standardized probit coefficients, standardized total effects, and standardized indirect effects were reported, with the bias-corrected 95% confidence intervals (CI) estimated using 1000 bootstrap samples. Path model fit indices were also reported, including comparative fit index (CFI; CFI > 0.90 indicating a good fit), root mean square error of approximation (RMSEA, RMSEA < 0.08 indicating an acceptable fit), and standardized root mean square residual (SRMR; SRMR < 0.08 indicating a good fit) [31,32]. All statistical analyses were conducted using SAS 9.2 (SAS Institute, Inc., Cary, NC, USA) and Mplus version 7.0 (Muthén and Muthén). The percentage of missing data of all relevant variables was less than 0.6, and observations with missing data were eliminated in the generalized linear mixed models and path models. All statistical tests were two-sided, and *p*-values less than 0.05 were considered statistically significant.

## 3. Results

The sample characteristics are shown in Table 1. A total of 13.5% of students reported living with others, and 13.9% reported below average HSS. The proportion of students who reported poor family relationships, classmate relations, and relationship with teachers, were 4.1%, 1.6%, and 2.4%, respectively. The mean IAT scores among the total students were 35.8 (SD: 12.8). A total of 0.6% and 0.3% of students reported frequent use of opioids and sedatives, respectively. The mean CES-D scores were 13.6 (SD: 8.7), and 6.7% of the students reported having depressive symptoms. There were significant differences in the CES-D scores among the variables of gender, living arrangement, HSS, academic performance, family relationships, classmate relations, relationships with teachers, opioid misuse, and sedative misuse (*p* < 0.001). Additionally, significant differences emerged between the students with and without depressive symptoms in the distribution of gender, living arrangement, HSS, academic performance, family relationships, classmate relations, relationships with teachers, IAT scores, opioid misuse, and sedative misuse (*p* < 0.001).

Without adjusting for other variables, Model 1 demonstrated that PIU, opioid misuse, and sedative misuse were respectively associated with the increase of depressive symptoms (*p* < 0.001). After adjusting for gender, living arrangement, HSS, academic performance, family relationships, classmate relations, and relationships with teachers, Model 2 showed that PIU was significantly associated with depressive symptoms (unstandardized β estimate = 0.26, 95% CI = 0.25–0.27), frequent use of opioids was positively related to depressive symptoms (unstandardized β estimate = 2.77, 95% CI = 1.90–3.63), and frequent use of sedatives was also related to the elevation of depressive symptoms (unstandardized β estimate = 4.45, 95% CI = 3.02–5.88). Additionally, Model 3 and Model 4 did not find any significant multiplicative interaction item between non-medical use of opioids/sedatives and PIU (Table 2).

As shown in Figure 1 and Table 3, the model including the mediator (opioid misuse or sedative misuse) showed that after adjusting for significant covariates, opioid misuse partially mediated the positive association between PIU and depressive symptoms, and the estimate of the standardized indirect effect was 0.003 (95% CI = 0.001–0.005). The obtained indices suggested that the model fit the data well: CFI = 0.92; RMSEA = 0.042, 95% CI = 0.030–0.055; SRMR = 0.061. However, the results also demonstrated that the adjusted standardized indirect effects of PIU on depressive symptoms through sedative misuse was not significant (*p* > 0.05), indicating that the relationship between PIU and depressive symptoms was not mediated by sedative misuse. The model fit indices indicated that the model fit the data satisfactorily: CFI = 0.89; RMSEA = 0.044, 95% CI = 0.032–0.047; SRMR = 0.052.

As shown in Figure 2 and Table 4, the alternative model incorporating the mediator (opioid misuse or sedative misuse) demonstrated that the adjusted indirect effects of depressive symptoms on PIU through opioid misuse were statistically significant (standardized β estimate = 0.002, 95% CI = 0.001–0.002), and the model fit indices were: CFI = 0.91; RMSEA = 0.042, 95% CI = 0.031–0.057; SRMR = 0.063. There were significant standardized indirect effects of depressive symptoms on PIU via sedative misuse (standardized β estimate = 0.001, 95% CI = 0.001–0.001), and the model indices suggested that the model fit the data well: CFI = 0.93; RMSEA = 0.040, 95% CI = 0.029–0.067; SRMR = 0.067. These results suggested that opioid misuse or sedative misuse partially mediated the association of depressive symptoms with PIU, respectively.

## 4. Discussion

To our knowledge, the present study is the first large-scale study to estimate the direct and indirect association between PIU, opioid or sedative misuse, and depressive symptoms among Chinese adolescents, and to investigate the potential mediating effects of opioid or sedative misuse on the association between PIU and depressive symptoms. This study first found that the mean CES-D scores for the adolescents were13.6 (SD: 8.7), and 6.7% reported having depressive symptoms. These results are parallel with our previous study conducted in 2014 [9], and slightly lower than that described in a prior review revealing a pooled prevalence of depressive symptoms of 24.3% (95% CI, 21.3–27.6%) among adolescents in mainland China [33]. The difference might be related to the variety in the definition of depressive symptoms (i.e., using different measurement scale and cutoff scores), and these results indicate that depressive symptoms among Chinese adolescents are a growing public health problem.

Consistent with previous evidence [9,11], the present study found that there were significant differences in the continuous and categorical variables of depressive symptoms among the groups of gender, living arrangement, HSS, academic performance, family relationships, classmate relations, relationships with teachers, opioid misuse, and sedative misuse. Compared with their corresponding groups, the mean CES-D scores were significantly higher in girls, students living in a single-parent family, students who reported below average HSS or academic performance, and those reporting poor relationships with family members, classmates, or teachers. These results are helpful to identify a profile of adolescents who are vulnerable to depressive symptoms, and particular attention should be paid to the groups with the negative characteristics mentioned above. Moreover, the covariate effects of these variables on the relationship between PIU and depressive symptoms should also be taken into consideration.

In the adjusted generalized linear mixed models without mediation, a positive relationship between depressive symptoms and PIU, opioid misuse, and sedative misuse was found, respectively. First, PIU may increase the risk of depressive symptoms due to adolescents with PIU showing poorer well-being, self-control, and self-esteem, which were reported to be positively associated with psychiatric disorders (including depressive symptoms) [34,35]. Similarly, Park et al. reported that PIU was positively associated with depressive symptoms in Korean adolescents [10]; Dalbudak et al. demonstrated that Internet addiction might increase vulnerability to depressive symptoms in students of Turkey [36]; and Tan et al. found that PIU was significantly related to an increased risk of depressive symptoms among adolescents in Shantou, China [11]. Moreover, the found significant association between opioid or sedative misuse and depressive symptoms might be related to the finding that these drugs may cause negative emotions and urgency, poor concentration, and sleeping problems, which are reported to be associated with an elevated risk of depressive symptoms [17,37]. Adolescents are especially vulnerable to the adverse effects of NMUPD given their still-developing brain [38].

Although there is evidence supporting the role of opioid or sedative misuse in the process through which PIU is related to depressive symptoms, no study before had confirmed the mediating effects of opioid or sedative misuse. Considering mediation analyses require exclusion of an interaction between the exposure and the mediator on the outcome [39], the interaction items between opioid/sedative misuse and PIU on depressive symptoms were first tested, and the results did not find any significant interaction effects. Furthermore, the path models of this study first demonstrated that the association between PIU and depressive symptoms remained significant when opioid or sedative misuse was incorporated as a mediator, and the association above was partially mediated through opioid misuse. These findings were consistent with the theory proposed by Caplan, which showed that PIU might be a reflection of maladaptive cognitions that lead to difficulties with behavioral impulse control and ultimately resulting in adverse outcomes associated with PIU (e.g., NMUPD and depressive symptoms) [18]. Moreover, a possible explanation for the mediating effects of opioid misuse might be that PIU and opioid use may have similarities in biological characteristics [3,15]. PIU may increase the risk of opioid misuse through impaired impulse control [40] and opioid misuse is one of the known factors associated with depressive symptoms [40]. In another aspect, Davis’s cognitive–behavioral theory proposed that depressive symptoms may also predispose individuals to develop maladaptive Internet-related cognitions and behaviors that can lead to PIU [19,20]. Moreover, due to the cross-sectional nature of data which may cause bias of the mediating estimates when mediation occurs over time, another path process order was also performed to test the potential the mediating effects opioid or sedative misuse on the association between depressive symptoms and PIU. The present study also demonstrated that opioid and sedative misuse significantly mediated the association of depressive symptoms with PIU, and these results may be related to the finding that adolescents with depressive symptoms may non-medically use opioid or sedatives as a coping strategy to release a negative mood [41], and substance abuse has been reported to be able to predict a high risk of PIU [42]. Taken together, the association between PIU, opioid or sedative misuse, and depressive symptoms might be complicated; for instance, PIU can lead to opioid or sedative misuse and depressive symptoms, which in turn may worsen the problematic Internet use problem among adolescents, and vice versa. The investigation of the mediating effects of opioid or sedative misuse can add evidence for the new understanding of the mechanism of the association between PIU and depressive symptoms.

Based on the findings of the current study, several recommendations for preventing PIU, NMUPD, and depressive symptoms among adolescents are listed below: (1) limiting adolescents’ exposure to the Internet for a long time (e.g., playing game); (2) increasing individuals’ awareness of the adverse effects of PIU, NMUPD, and depressive symptoms; (3) providing professional health services (e.g., health services provided by clinicians or social workers in the schools or communities) to students who have been involved in PIU, NMUPD, and depressive symptoms. If possible, concomitant treatment for PIU, NMUPD, and depressive symptoms should be considered; and (4) developing a long-term surveillance system to monitor the health-risky behaviors among adolescents in China.

There are several limitations related to this study. First, the use of self-report questionnaires in this study may result in the underestimate of some sensitive data among adolescents (e.g., PIU or NMUPD) for social desirability. Second, due to the cross-sectional nature of the study design, findings should be interpreted with caution, especially regarding the longitudinal indirect and direct effects. Third, because of the school-based study design, the study sample only included students at school, while PIU, NMUPD, or depressive symptoms may be more common among adolescents who were absent from schools. Fourth, PIU and depressive symptoms were measured by the IAT and CES-D scales. Although this measurement of PIU or depressive symptoms has been validated and widely used in previous studies, the answers may still be subjectively biased. Fifth, Axis I comorbidities (such as anxiety or bipolar disorder), which may have an effect on depressive symptoms, were not taken into account in the present study. Despite these limitations, the strength of this study is that it uses a large-scale sample of Chinese adolescents to extend the prior evidence about the association between PIU, opioid or sedative misuse, and depressive symptoms through investigating the mediating effects of opioid or sedative misuse.

## 5. Conclusions

In conclusion, the present study found that in the models without mediation, PIU and opioid/sedative misuse were positively related to an increased risk of depressive symptoms among adolescents, respectively. Moreover, two path models were conducted to estimate the potential mechanism of the association between PIU, opioid or sedative misuse, and depressive symptoms. The results indicated that the association above might be complex and transactional, and that PIU may elevate the risk of opioid or sedative misuse and depressive symptoms, which in turn could worsen the situation of PIU and vice versa. These study findings have some important implications. Multidisciplinary health intervention programs to prevent adolescents from getting involved in PIU and NMPUD are recommended to be provided, and the concomitant or complex transactional association between PIU, opioid or sedative misuse, and depressive symptoms should be taken into consideration by adolescents, their families and teachers, as well as professional health providers.

## Figures and Tables

**Figure 1 ijerph-17-00774-f001:**
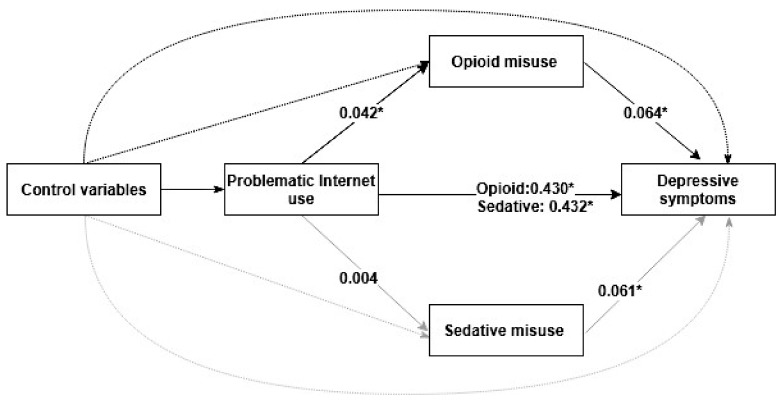
The mediating effects of opioid or sedative misuse on the association between problematic Internet use and depressive symptoms. The solid lines mean the effects between the independent variable, mediator variables, and dependent variable. The dashed lines mean the effects of control variables (gender, living arrangement, HSS, academic performance, family relationships, classmate relations, and relationships with teachers) on the independent variable, mediator variables, and dependent variable.

**Figure 2 ijerph-17-00774-f002:**
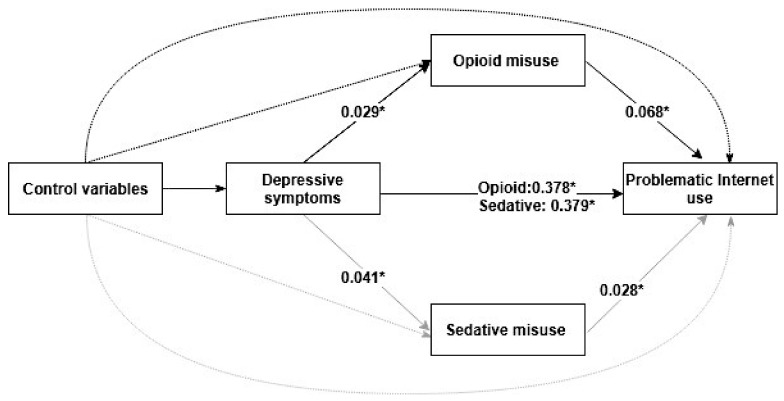
The mediating effects of opioid or sedative misuse on the association between depressive symptoms and problematic Internet use. The solid lines mean the effects between the independent variable, mediator variables, and dependent variable. The dashed lines mean the effects of control variables (gender, living arrangement, HSS, academic performance, family relationships, classmate relations, and relationships with teachers) on the independent variable, mediator variables, and dependent variable.

**Table 1 ijerph-17-00774-t001:** Sample characteristics stratified by depressive symptoms among 24,345 adolescents.

Variable	Total	CES-D Scores, Mean (SD)	*p*-Value *	Depressive Symptoms		*p*-Value *
				Yes	No	
Total	24,345 (100)	13.6 (8.7)		1631 (6.7)	22,714 (93.3)	
Gender						
Boys	12,526 (51.5)	12.9 (8.6)	<0.001	731 (45.0)	11,795 (52.1)	<0.001
Girls	11,732 (48.2)	14.4 (8.8)		892 (55.0)	10,840 (47.9)	
Missing data	87 (0.4)					
Living arrangement						
Living in two-parent family	18,094 (74.3)	13.3 (8.6)	<0.001	1125 (69.1)	16,969 (74.9)	<0.001
Living in a single-parent family	2905 (11.9)	15.0 (9.4)		259 (15.9)	2646 (11.7)	
Living with others	3281 (13.5)	14.3 (8.8)		243 (14.9)	3038 (13.4)	
Missing data	65 (0.3)					
HSS						
Above average	6942 (28.5)	12.1 (8.1)	<0.001	315 (19.3)	6627 (29.3)	<0.001
Average	13,944 (57.3)	13.8 (8.6)		928 (57.0)	13,016 (57.5)	
Below average	3388 (13.9)	16.3 (9.8)		385 (23.6)	3003 (13.3)	
Missing data	71 (0.3)					
Academic performance						
Above average	9195 (37.8)	12.3 (8.5))	<0.001	508 (31.2)	8687 (38.5)	<0.001
Average	7576 (31.1)	13.5 (8.2)		434 (26.7)	7142 (31.6)	
Below average	7448 (30.6)	15.5 (9.3)		685 (42.1)	6763 (29.9)	
Missing data	126 (0.5)					
Family relationships						
Good	19,899 (81.7)	12.6 (8.0)	<0.001	946 (58.1)	18,953 (83.8)	<0.001
Average	3362 (13.8)	17.6 (9.8)		429 (26.3)	2933 (13.0)	
Poor	986 (4.1)	21.5 (11.9)		254 (15.6)	732 (3.2)	
Missing data	98 (0.4)					
Classmate relations						
Good	19,561 (80.3)	12.5 (7.9)	<0.001	899 (55.3)	18,662 (82.6)	<0.001
Average	4274 (17.6)	17.9 (9.7)		580 (35.7)	3694 (16.4)	
Poor	381 (1.6)	26.3 (13.7)		146 (9.0)	235 (1.0)	
Missing data	129 (0.5)					
Relationship with teachers						
Good	15,695 (64.5)	12.1 (7.9)	<0.001	706 (43.6)	14,989 (66.6)	<0.001
Average	7844 (32.2)	16.2 (9.2)		790 (48.8)	7054 (31.4)	
Poor	576 (2.4)	21.2 (12.6)		124 (7.7)	452 (2.0)	
Missing data	230 (0.9)					
IAT scores, Mean (SD)	35.8 (12.8)	NA		49.3 (16.5)	34.8 (11.9)	<0.001
Opioid misuse						
Abstainers	23,822 (97.9)	13.6 (8.7)	<0.001	1559 (95.6)	22,263 (98.0)	<0.001
Experimenters	384 (1.6)	17.3 (10.0)		49 (3.0)	335 (1.5)	
Frequent users	139 (0.6)	19.2 (9.8)		23 (1.4)	116 (0.5)	
Sedative misuse						
Abstainers	24,095 (99.0)	13.6 (8.7)	<0.001	1588 (97.4)	22,507 (99.1)	<0.001
Experimenters	188 (0.8)	17.4 (9.1)		23 (1.4)	165 (0.7)	
Frequent users	62 (0.3)	23.8 (14.6)		20 (1.2)	42 (0.2)	

Note: * *t*-tests or one-way ANOVA tests were performed to test the differences in CES-D score, and Chi-square tests or *t*-tests were used to compare the differences in adolescents with and without depressive symptoms; HSS—household socioeconomic status; IAT—internet addiction test; SD–standard deviation; CES-D—Center for Epidemiologic Studies Depression; NA—not applicable or no data available.

**Table 2 ijerph-17-00774-t002:** Association between problematic Internet use, opioid misuse, sedative misuse, and depressive symptoms among adolescents.

Variable	CES-D scores							
	Model 1		Model 2		Model 3		Model 4	
	β Estimate ^#^ (95% CI)	*p*-Value	β Estimate ^#^ (95% CI)	*p*-Value	β Estimate ^#^ (95% CI)	*p*-Value	β Estimate ^#^ (95% CI)	*p*-Value
Problematic Internet use (1-score increase)	0.30 (0.29–0.31)	<0.001	0.26 (0.25–0.27)	<0.001	0.26 (0.25–0.27)	<0.001	0.26 (0.25–0.27)	<0.001
Opioid misuse (Ref. = Abstainers)								
Experimenters	3.75 (2.82–4.68)	<0.001	2.77 (1.90–3.63)	<0.001	2.42 (0.19–4.65)	0.034	NA	
Frequent users	5.65 (4.11–7.12)	<0.001	4.45 (3.02–5.88)	<0.001	3.95(−0.32–8.22)	0.071	NA	
Sedative misuse (Ref.= Abstainers)								
Experimenters	3.85 (2.54–5.17)	<0.001	2.86 (1.63–4.09)	<0.001	NA		2.53 (−0.86–5.91)	0.144
Frequent users	10.26 (8.03–12.48)	<0.001	7.18 (5.09–9.26)	<0.001	NA		10.85 (5.15–16.56)	<0.001
Interaction item (opioid misuse)								
Experimenters * Problematic Internet use	NA		NA		−0.02 (−0.07–0.03)	0.502	NA	
Frequent users * Problematic Internet use	NA		NA		−0.04 (−0.13–0.05)	0.387	NA	
Interaction item (sedative misuse)								
Experimenters * Problematic Internet use	NA		NA		NA		−0.01 (−0.09–0.08)	0.884
Frequent users * Problematic Internet use	NA		NA		NA		−0.11 (−0.22–0.01)	0.086

Note: ^#^ Unstandardized β coefficient. Model 1: Unadjusted generalized linear mixed models. Model 2: The multivariable generalized linear mixed models were adjusted for gender, living arrangement, HSS, academic performance, family relationships, classmate relations, and relationships with teachers. Model 3: The multivariable generalized linear mixed models simultaneously incorporated the interaction item for opioid misuse and problematic Internet use along with the main effect terms for each, and were adjusted for gender, living arrangement, HSS, academic performance, family relationships, classmate relations, and relationships with teachers. Model 4: The multivariable generalized linear mixed models simultaneously incorporated the interaction item for sedative misuse and problematic Internet use along with the main effect terms for each, and were adjusted for gender, living arrangement, HSS, academic performance, family relationships, classmate relations, and relationships with teachers; Ref.—reference; CES-D—Center for Epidemiologic Studies Depression; NA—not applicable or no data available. * The multiplicative interaction between the two items.

**Table 3 ijerph-17-00774-t003:** Path analysis showing the effects of problematic Internet use and opioid or sedative misuse on depressive symptoms.

Variable.	Symbol	CES-D Scores	
		Unadjusted Model	Adjusted Model *
		Standardized β Estimate (95% CI)	Standardized β Estimate (95% CI)
Problematic Internet use » Depressive symptoms	Predictor » Outcome	0.437 (0.425–0.449)	0.430 (0.418–0.442)
Opioid misuse » Depressive symptoms	Mediator » Outcome	0.072 (0.058–0.086)	0.064 (0.048–0.080)
Problematic internet use » Opioid misuse	Predictor » Mediator	0.045 (0.033–0.057)	0.042 (0.030–0.054)
Standardized effect			
Indirect		0.003 (0.001–0.005)	0.003 (0.001–0.005)
Total		0.441 (0.429–0.453)	0.430 (0.418–0.442)
Problematic Internet use » Depressive symptoms	Predictor » Outcome	0.4395 (0.4280–0.4518)	0.4318 (0.4202–0.4439)
Sedative misuse » Depressive symptoms	Mediator » Outcome	0.072 (0.058–0.086)	0.061 (0.045–0.077)
Problematic internet use » Sedative misuse	Predictor » Mediator	0.007 (−0.005–0.019)	0.004 (−0.008–0.016)
Standardized effect			
Indirect		0.0005 (−0.002–0.0006)	0.0002 (−0.0018–0.0003)
Total		0.440 (0.428–0.452)	0.432 (0.420–0.444)

Note: * The path models were adjusted for gender, living arrangement, HSS, academic performance, family relationships, classmate relations, and relationships with teachers; CES-D—Center for Epidemiologic Studies Depression.

**Table 4 ijerph-17-00774-t004:** Path analysis showing the effects of depressive symptoms and opioid or sedative misuse on problematic Internet use.

Variable	Symbol	Problematic Internet Use	
		Unadjusted Model	Adjusted Model *
		Standardized β Estimate (95% CI)	Standardized β Estimate (95% CI)
CES-D scores » Problematic Internet use	Predictor » Outcome	0.438 (0.426–0.450)	0.378 (0.366–0.390)
Opioid misuse » Problematic Internet use	Mediator » Outcome	0.076 (0.062–0.090)	0.068 (0.054–0.082)
CES-D scores » Opioid misuse	Predictor » Mediator	0.038 (0.026–0.050)	0.029 (0.017–0.041)
Standardized effect			
Indirect		0.003 (0.001–0.005)	0.002 (0.001–0.002)
Total		0.441 (0.429–0.453)	0.380 (0.368–0.392)
CES-D scores » Problematic Internet use	Predictor » Outcome	0.438 (0.426–0.450)	0.379 (0.367–0.391)
Sedative misuse » Problematic Internet use	Mediator » Outcome	0.039 (0.025–0.053)	0.028 (0.014–0.042)
CES-D scores » Sedative misuse	Predictor » Mediator	0.056 (0.044–0.068)	0.041 (0.029–0.053)
Standardized effect			
Indirect		0.002 (0.001–0.002)	0.001 (0.001–0.001)
Total		0.440 (0.429–0.453)	0.380 (0.368–0.392)

Note: * The path models were adjusted for gender, living arrangement, HSS, academic performance, family relationships, classmate relations, and relationships with teachers; CES-D—Center for Epidemiologic Studies Depression.

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
