# Peer review of "Problematic Internet Use, Non-Medical Use of Prescription Drugs, and Depressive Symptoms among Adolescents: A Large-Scale Study in China"

_ijerph, 2020, doi:10.3390/ijerph17030774_

Round 1

Reviewer 1 Report

1. In the Study design and participants section, the descriptions read like the authors conducted the SCAHS (e.g., the authors wrote "line 95 we randomly selected two representative cities from each stratification..."; "line 100 and our research assistants administered these questionnaires during a regular class period"). If the authors did conduct the SCAHS, please explicitly mention this in this section. If the authors did not conduct the SCAHS but obtained the data from any organization, please also clearly indicate how and where (web link is fine) they obtained the data.
2. Following the previous comment, the sentence shown in line 88 "SCAHS performed large-scale cross-sectional surveys every two years since 2007 and conducted a longitudinal study from 2009 to 2012" is unclear to me. Specifically, does this mean that since 2007, the survey was conducted every two years and every two years the survey questions changed (or the samples changed)? Then, by 2009 and 2012, the survey questions were kept the same using the same sample. But, the assessed timing read weird because for every two years, the longitudinal surveys can be conducted by 2009 and 2011 only. Why did the authors mentioned 2012?
3. line 113, shouldn't the (23) be [23]?
4. In the abstract results and Results section in the main text, it is unclear whether the beta estimates indicates standardized or unstandardized coefficients. I think they are unstandardized coefficients. Also, I am confused that why the authors provided beta estimates but not odds ratio? The authors used multilevel logistic regression; then, it is better to report odds ratio rather than to report beta.
5. Following my previous comment, I cannot understand why the authors want to use logistic regression instead of linear regression. The authors have the information of continuous score on CES-D. From the statistical perspective, it will be always better to use all the information rather than cutting some information. That says, converting the continuous score of CES-D into two ends (having depression vs. not having depression) usually lose some information for analyses. Additionally, the percentage of those who reported depressive symptoms is not large (6.7%); thus, the unbalanced ratio between samples with and without depressive symptoms is likely to jeopardize the analyzed results.
6. Tables 3 and 4 are hard to read. Suggest (1) to add a horizontal line between the raw showing Total and the raw showing problematic internet use. Thus, readers can easily understand that there are two mediation parts in the model; (2) to add the final outcome for both predictor and mediator. That is, problematic internet useàdepression; opioid misuseàdepression; predictoràoutcome; mediatoràoutcome. I think that this will make the presentations of problematic internet useàopioid misuse and predictoràmediator clearer; (3) to change the column title “Model 1” and “Model 2”; I think that using “unadjusted model” and “adjusted model” will be better.
7. Following my previous comment, I wonder why the indirect standardized effect has a 0 in Table 3. Also, why the direct standardized effect is an equal value to the standardized estimate of problematic internet use to depression (i.e., 0.440). Given that sedative misuse to depression has a value of 0.072 and the problematic internet use to sedative misuse has a value of 0.007; there should have an indirect standardized effect of 0.00504. Although the value is very small, it should not be stated as 0. Also, there should be a confidence interval for the indirect standardized effect here.
8. For figures 1 and 2, please provide clear footnote to describe the meaning of dashed and sold lines.

Author Response

Response to Reviewer 1 Comments

Point 1: In the Study design and participants section, the descriptions read like the authors conducted the SCAHS (e.g., the authors wrote "line 95 we randomly selected two representative cities from each stratification..."; "line 100 and our research assistants administered these questionnaires during a regular class period"). If the authors did conduct the SCAHS, please explicitly mention this in this section. If the authors did not conduct the SCAHS but obtained the data from any organization, please also clearly indicate how and where (web link is fine) they obtained the data.

Response 1: We truly appreciate your time in reviewing our article. First, our group conducted the SCAHS, and we have published a series of articles before [1-4]. Second, as suggested, we have added this information to the revised manuscript (please see line 101).

Point 2: Following the previous comment, the sentence shown in line 88 "SCAHS performed large-scale cross-sectional surveys every two years since 2007 and conducted a longitudinal study from 2009 to 2012" is unclear to me. Specifically, does this mean that since 2007, the survey was conducted every two years and every two years the survey questions changed (or the samples changed)? Then, by 2009 and 2012, the survey questions were kept the same using the same sample. But, the assessed timing read weird because for every two years, the longitudinal surveys can be conducted by 2009 and 2011 only. Why did the authors mentioned 2012?

Response 2: We apologize for the misunderstanding caused by the unclear text. First, the SCAHS conducted a series of cross-sectional surveys every two years since 2007 (the samples changed) [1-3]. Additionally, the SCAHS also conducted a longitudinal study from 2009 to 2012 (in a small sample in Guangzhou city) [5]. Second, to make this paragraph more clear, we have removed the sentence about the longitudinal study (please see lines 101-103).

Point 3: line 113, shouldn't the (23) be [23]?

Response 3: Thank you for carefully and patiently reviewing our manuscript. We have revised the reference style (please see line 115).

Point 4: In the abstract results and Results section in the main text, it is unclear whether the beta estimates indicates standardized or unstandardized coefficients. I think they are unstandardized coefficients. Also, I am confused that why the authors provided beta estimates but not odds ratio? The authors used multilevel logistic regression; then, it is better to report odds ratio rather than to report beta.

Response 4: Thank you for noticing this. We apologize for the expression error in the original manuscript. Actually, in this study, due to the outcome variable (the CES-D scores) was a continuous variable, generalized linear mixed models were performed, and the unstandardized coefficients were reported. According to your suggestion, we have revised these sections to make it more clear (please see lines 19-23 & lines 165-168 & lines 205-208).

Point 5: Following my previous comment, I cannot understand why the authors want to use logistic regression instead of linear regression. The authors have the information of continuous score on CES-D. From the statistical perspective, it will be always better to use all the information rather than cutting some information. That says, converting the continuous score of CES-D into two ends (having depression vs. not having depression) usually lose some information for analyses. Additionally, the percentage of those who reported depressive symptoms is not large (6.7%); thus, the unbalanced ratio between samples with and without depressive symptoms is likely to jeopardize the analyzed results.

Response 5: Thank you for your kind suggestion. We apologize again for the expression error in the original manuscript (please see the Response of Point 4). In the present study, the outcome variable of depressive symptoms was a continuous variable (the CES-D scores), and then generalized linear mixed models were performed for the multi-stage sampling design. We have revised these sections to make it more clear (please see lines 165-168).

Point 6: Tables 3 and 4 are hard to read. Suggest (1) to add a horizontal line between the raw showing Total and the raw showing problematic internet use. Thus, readers can easily understand that there are two mediation parts in the model; (2) to add the final outcome for both predictor and mediator. That is, problematic internet use à depression; opioid misuse à depression; predictor à outcome; mediator à outcome. I think that this will make the presentations of problematic internet use à opioid misuse and predictor à mediator clearer;

(3) to change the column title “Model 1” and “Model 2”; I think that using “unadjusted model” and “adjusted model” will be better.

Response 6: Thank you for your kind suggestion. According to your suggestion, we have revised the tables to make it more clear (please see the revised Table 3 and Table 4).

Point 7: Following my previous comment, I wonder why the indirect standardized effect has a 0 in Table 3. Also, why the direct standardized effect is an equal value to the standardized estimate of problematic internet use to depression (i.e., 0.440). Given that sedative misuse to depression has a value of 0.072 and the problematic internet use to sedative misuse has a value of 0.007; there should have an indirect standardized effect of 0.00504. Although the value is very small, it should not be stated as 0. Also, there should be a confidence interval for the indirect standardized effect here.

Response 7: Thank you for your kind suggestion. We apologize for this expression error (the rounding off of the numbers) in Table 3. According to your suggestion, we have revised Table 3 to make it more clear.

Point 8: For figures 1 and 2, please provide clear footnote to describe the meaning of dashed and sold lines.

Response 8: According to your suggestion, we have provided the footnotes to describe the meaning of solid and dashed lines (please see the revised Figure 1 and Figure 2).

References:

Guo, L.; Xu, Y.; Deng, J.; Huang, J.; Huang, G.; Gao, X.; Li, P.; Wu, H.; Pan, S.; Zhang, W.H., et al. Association between sleep duration, suicidal ideation, and suicidal attempts among Chinese adolescents: The moderating role of depressive symptoms. J Affect Disord 2017, 208, 355-362, doi:10.1016/j.jad.2016.10.004. Guo, L.; Xu, Y.; Deng, J.; Gao, X.; Huang, G.; Huang, J.; Deng, X.; Zhang, W.H.; Lu, C. Associations between childhood maltreatment and non-medical use of prescription drugs among Chinese adolescents. Addiction 2017, 112, 1600-1609, doi:10.1111/add.13850. Guo, L.; Wang, W.; Gao, X.; Huang, G.; Li, P.; Lu, C. Associations of Childhood Maltreatment with Single and Multiple Suicide Attempts among Older Chinese Adolescents. J Pediatr 2018, 196, 244-250, doi:10.1016/j.jpeds.2018.01.032. Guo, L.; Wang, W.; Wang, T.; Li, W.; Gong, M.; Zhang, S.; Zhang, W.H.; Lu, C. Association of emotional and behavioral problems with single and multiple suicide attempts among Chinese adolescents: Modulated by academic performance. J Affect Disord 2019, 258, 25-32, doi:10.1016/j.jad.2019.07.085. Guo, L.; Xu, Y.; Deng, J.; Huang, J.; Huang, G.; Gao, X.; Wu, H.; Pan, S.; Zhang, W.H.; Lu, C. Association Between Nonmedical Use of Prescription Drugs and Suicidal Behavior Among Adolescents. Jama Pediatr 2016, 170, 971-978, doi:10.1001/jamapediatrics.2016.1802.

Reviewer 2 Report

- It is recommended to add part of the conclusions in the abstract.

- the idea underlying the study  is improvable

-In the introduction, please, make sure that the paragraphs are connected.

- clarify alpha indexes of original and this study scales.

-clarify the procedure for selecting participants (inclusion and exclusion criteria)

-personal variables that may influence mood have not been taken into account. justify it

-The justified study needs to be justified

- justify the study in the adolescent population (7th-12th grade)

Author Response

Response to Reviewer 2 Comments

Point 1: It is recommended to add part of the conclusions in the abstract.

Response 1: Thank you for carefully and patiently reviewing our manuscript. According to your suggestion, we have revised the abstract section (please see lines 15-31). Considering the word limit (a total of about 200 words maximum), we use concise words to discuss key messages.

Point 2: the idea underlying the study is improvable.

Response 2: Thank you for your kind suggestion. In the present study, the first paragraph describes that Internet use has proliferated dramatically over the last decades and has become an internal part of contemporary life [1], and excessive or maladaptive use of the Internet has been termed as “problematic Internet use (PIU)” or “Internet addiction”, which has been reported to be related to a series of psychosocial impairments and substance use problems (e.g., depression and drug use) [2,3]. Additionally, adolescence represents a vulnerable developmental stage between childhood to adulthood, which is characterized by increased levels of novelty seeking and exploration along with a wide range of risk-taking behaviors or mental health problems [4]. The report of the World Health Organization (WHO) showed that adolescent PIU had been a public health concern worldwide, and China is no exception [5]. Considering the brain undergoes rapid development during adolescence, adolescents involved in PIU may be more vulnerable to psychological disorders, such as depressive symptoms. Then, the second paragraph describes that depressive symptoms are one of the most common mental disorders among adolescents. Although the association between PIU and depressive symptoms has been documented in prior studies [6,7], the influence path of PIU on depressive symptoms is still unclear (please see the second paragraph). Next, the third paragraph says that previous studies demonstrated that PIU was reported to be associated with substance use, and PIU and substance use may share similar biological characteristics (e.g., similar vulnerable brain regions) [8,9]. Moreover, individuals with substance use may affect the brain function, as well as the ability of self-regulation, and then lead to the development of depressive symptoms [10,11]. Therefore, NMUPD may play a mediator role in the association between PIU and depressive symptoms (please see the third paragraph).

Point 3: In the introduction, please, make sure that the paragraphs are connected.

Response 3: We thank the reviewer for this comment. In the Introduction section, we have clearly described the background of this study (please see the response of Point 2). Additionally, as suggested, we have revised the Introduction section to make it more clear (please see lines 68-87).

Point 4: clarify alpha indexes of original and this study scales.

Response 4: Thank you for your kind suggestion. The Cronbach’s alpha of the scales in different study sample might be various, and in the original manuscript, we have demonstrated that in the present study, the Cronbach’s alpha for this CES-D scale was 0.88 and the Cronbach’s alpha for IAT was 0.91 (please see line 128 & line 135). Moreover, according to your suggestion, we have revised this section to make it more clear (please see lines 126-129 & lines133-134), and Young’s original study did not report the Cronbach’s alpha.

Point 5: clarify the procedure for selecting participants (inclusion and exclusion criteria).

Response 5: Thank you for your question. In the original manuscript, we have clearly demonstrated that this study was a cross-sectional study, and all available students were invited to participate in this study voluntarily. Moreover, all participants were informed about the study, and all provided written informed consents. If the participant was under 18 years of age, a written informed consent letter was obtained from one of the student’s parents (or legal guardian) (please see lines 112-123).

Point 6: personal variables that may influence mood have not been taken into account. justify it.

Response 6: Thank you for your kind suggestion. First, in this study, PIU, NMUPD, gender, age, living arrangement, household socioeconomic status, academic performance, family relationships, classmate relations, and relationships with teachers were taken into account, and these variables could be considered as personal variables (please see lines 125-158). Second, as suggested, we have added a limitation to the revised manuscript (please see lines 350-352).

Point 7: The justified study needs to be justified.

Response 7: We thank the reviewer for this comment. We have revised this manuscript to make it more clear.

Point 8: justify the study in the adolescent population (7th-12th grade).

Response 8: Thank you for your kind suggestion. First, adolescence is often described as occurring between 13 and 18 years of age, which is roughly the period of high school (7th-12th grade) for much of the world [12]. As suggested, we have added this information to the revised manuscript (please see lines 109-110). Second, In the original manuscript, we have clearly demonstrated that the mean age of students was 15.2 ± 1.8 years (please see line 118).

References:

Lin, M.P.; Wu, J.Y.; Chen, C.J.; You, J. Positive outcome expectancy mediates the relationship between social influence and Internet addiction among senior high-school students. J Behav Addict 2018, 1-9, doi:10.1556/2006.7.2018.56. An, J.; Sun, Y.; Wan, Y.; Chen, J.; Wang, X.; Tao, F. Associations between problematic internet use and adolescents' physical and psychological symptoms: possible role of sleep quality. J Addict Med 2014, 8, 282-287, doi:10.1097/ADM.0000000000000026. Rucker, J.; Akre, C.; Berchtold, A.; Suris, J.C. Problematic Internet use is associated with substance use in young adolescents. Acta Paediatr 2015, 104, 504-507, doi:10.1111/apa.12971. Tymula, A.; Rosenberg, B.L.; Roy, A.K.; Ruderman, L.; Manson, K.; Glimcher, P.W.; Levy, I. Adolescents' risk-taking behavior is driven by tolerance to ambiguity. Proc Natl Acad Sci U S A 2012, 109, 17135-17140, doi:10.1073/pnas.1207144109. World Health Organization. Public Health Implications of Excessive Use of the Internet, Computers, Smartphones and Similar Electronic Devices Meeting report. In Main Meeting Hall, Foundation for Promotion of Cancer Research National Cancer Research Centre, Tokyo, Japan, 2015. Park, S.; Hong, K.E.; Park, E.J.; Ha, K.S.; Yoo, H.J. The association between problematic internet use and depression, suicidal ideation and bipolar disorder symptoms in Korean adolescents. Aust N Z J Psychiatry 2013, 47, 153-159, doi:10.1177/0004867412463613. Tan, Y.; Chen, Y.; Lu, Y.; Li, L. Exploring Associations between Problematic Internet Use, Depressive Symptoms and  Sleep Disturbance among Southern Chinese Adolescents. Int J Environ Res Public Health 2016, 13, doi:10.3390/ijerph13030313. Rucker, J.; Akre, C.; Berchtold, A.; Suris, J.C. Problematic Internet use is associated with substance use in young adolescents. Acta Paediatr 2015, 104, 504-507, doi:10.1111/apa.12971. Han, D.H.; Hwang, J.W.; Renshaw, P.F. Bupropion sustained release treatment decreases craving for video games and cue-induced brain activity in patients with Internet video game addiction. Exp Clin Psychopharmacol 2010, 18, 297-304, doi:10.1037/a0020023. Brand, M.; Young, K.S.; Laier, C. Prefrontal control and internet addiction: a theoretical model and review of neuropsychological and neuroimaging findings. Front Hum Neurosci 2014, 8, 375, doi:10.3389/fnhum.2014.00375. Quello, S.B.; Brady, K.T.; Sonne, S.C. Mood disorders and substance use disorder: a complex comorbidity. Sci Pract Perspect 2005, 3, 13-21. Crockett, L.J.; Beal, S.J. The life course in the making: gender and the development of adolescents' expected timing of adult role transitions. Dev Psychol 2012, 48, 1727-1738, doi:10.1037/a0027538.

Round 2

Reviewer 1 Report

I thank the authors to clearly responded to my previous comments. Although the manuscript is much improved, there are some unclear parts need to be further clarified.

1. line164. Per answer from the authors that they used generalized linear mixed models with the dependent variable as a continuous variable (i.e., CES-D), why did the authors have standardized probit coefficients?
2. The authors have reported model fit indices in the Results. However, they did not describe the meaning of these fit indices in the Methods section. Please describe the fit indices you used and your proposed cutoffs for the fit indices.
3. ll198-209. Please check the CFI values. They are too small and I doubt that the authors mistakenly add one 0 after the decimal point. 

Author Response

Response to Reviewer 1 Comments

Point 1: line164. Per answer from the authors that they used generalized linear mixed models with the dependent variable as a continuous variable (i.e., CES-D), why did the authors have standardized probit coefficients?

Response 1: We truly appreciate your time in reviewing our manuscript, and we apologize for the misunderstanding caused by the unclear texts. First, in the present study, we have demonstrated that univariable generalized linear mixed models were first conducted to estimate the potential association of PIU or NMUPD with depressive symptoms, and multivariable generalized linear mixed models in which variables that were significant at 0.10 level in univariate analyses or widely reported in the literature were simultaneously incorporated were performed to test the independent association of PIU or NMUPD with depressive symptoms (please see lines 149-155). Then, unstandardized β coefficients were reported. Second, in this study, we also conducted path models to estimate the mediating effects of NMUPD on the association between PIU and depressive symptoms. In the original manuscript, we have demonstrated that path models utilizing the maximum likelihood approach were conducted to assess the mediating effects of opioid or sedative misuse on the association between PIU and depressive symptoms. We first ordered the variables as follows: PIU—opioid or sedative misuse—depressive symptoms, and then an alternative model with a different order of the variables were assessed: depressive symptoms—opioid or sedative misuse—PIU. Due to the variables of CES-D scores and IAT scores were continuous variables, and the measures of opioid and sedative misuse were categorized, standardized probit coefficients, standardized total effects, and standardized indirect effects were reported, with the bias-corrected 95% confidence intervals (CI) estimated using 1,000 bootstrap samples (please see lines 158-167). Moreover, this statistical method and the corresponding estimates have been utilized in our previous studies [1,2]. Third, according to your suggestion, we have revised the statistical analysis section to make it more clear (please see lines 150-151).

Point 2: The authors have reported model fit indices in the Results. However, they did not describe the meaning of these fit indices in the Methods section. Please describe the fit indices you used and your proposed cutoffs for the fit indices.

Response 2: Thank you for your constructive suggestions. We have added this information to the revised manuscript (please see lines 167-170).

Point 3: ll198-209. Please check the CFI values. They are too small and I doubt that the authors mistakenly add one 0 after the decimal point.

Response 3: Thank you for carefully and patiently reviewing our manuscript, and we apologize for these mistakes in the original manuscript. As suggested, we have re-checked the manuscript and re-analyzed the data, and these CFI values have been corrected (please see line 202, 206, 210, or 213).

References:

Guo, L.; Xu, Y.; Deng, J.; Huang, J.; Huang, G.; Gao, X.; Wu, H.; Pan, S.; Zhang, W.H.; Lu, C. Association Between Nonmedical Use of Prescription Drugs and Suicidal Behavior Among Adolescents. Jama Pediatr 2016, 170, 971-978, doi:10.1001/jamapediatrics.2016.1802. Guo, L.; Luo, M.; Wang, W.X.; Huang, G.L.; Xu, Y.; Gao, X.; Lu, C.Y.; Zhang, W.H. Association between problematic Internet use, sleep disturbance, and suicidal behavior in Chinese adolescents. J Behav Addict 2018, 1-11, doi:10.1556/2006.7.2018.115.

Reviewer 2 Report

- the authors have modified the revisions raised by the reviewer.